# Metastatic Risk Factors Associated with Class 1A Uveal Melanoma Patients

**DOI:** 10.3390/cancers13133292

**Published:** 2021-06-30

**Authors:** Alexej Ballhausen, Elizabeth Urias, Stephen K. Gruschkus, Michelle Williams, Maura S. Glover, Yong Qin, Dan S. Gombos, Sapna P. Patel

**Affiliations:** 1Department of Melanoma Medical Oncology, Division of Cancer Medicine, The University of Texas MD Anderson Cancer Center, Houston, TX 77030, USA; alexej.ballhausen@charite.de (A.B.); msglover@mdanderson.org (M.S.G.); 2Medical Department, Division of Hematology, Oncology and Tumor Immunology, Charité-Universitätsmedizin Berlin, 13353 Berlin, Germany; 3Department of Head and Neck Surgery, Section of Ophthalmology, The University of Texas MD Anderson Cancer Center, Houston, TX 77030, USA; elizannurias@gmail.com (E.U.); dgombos@mdanderson.org (D.S.G.); 4Division of Biostatistics, The University of Texas MD Anderson Cancer Center, Houston, TX 77030, USA; sgruschkus@gmail.com; 5Department of Pathology, Division of Pathology and Laboratory Medicine, The University of Texas MD Anderson Cancer Center, Houston, TX 77030, USA; MDWillia@mdanderson.org; 6Department of Pharmaceutical Sciences, The University of Texas at El Paso, El Paso, TX 79902, USA; yqin@utep.edu

**Keywords:** uveal melanoma, metastasis, GEP, Class 1A, risk factors

## Abstract

**Simple Summary:**

Uveal melanoma (UM), patients with Class 1A gene expression profiling (GEP), have a lower metastatic risk (2% at 5 years) compared to Class 1B or Class 2 patients. However, further risk stratification could help to adapt follow-up intervals in Class 1A UM patients according to their metastatic risk. Our single-center IRB-approved retrospective case series review of 73 Class 1A UM patients aimed to identify risk factors associated with metastasis development and overall survival. We show that Class 1A UM patients with stage III disease and/or large COMS size are at elevated risk for metastasis. Combined clinical decision-making utilizing AJCC stage and COMS size could have a significant clinical impact by improving risk stratification and adapting follow-up intervals in Class 1A UM patients.

**Abstract:**

In uveal melanoma (UM), gene expression profiling (GEP) is commonly used to classify metastatic risk into three groups (Class 1A, 1B, and 2). Class 1A patients have a lower metastatic risk of 2% at 5 years compared to other groups. We aimed to describe clinical features associated with the development of metastasis in this low-risk group. This single-center IRB-approved retrospective case series review included all UM patients between February 2006 and March 2019 with an archived or fresh specimen classified as Class 1A. Cox regression and receiver operating characteristics analyses were used to identify factors associated with metastasis development and OS. Among 73 UM patients with Class 1A, the 5-year cumulative incidence of local recurrence and distant metastasis was 4.2% and 17.0%, respectively. Stage III disease (HR 20.7; 95% confidence interval (95% CI) 1.4–300.6; *p*
*=* 0.0264) was found to be independently associated with metastatic recurrence, while primary therapy was associated with OS (enucleation vs. brachytherapy, HR 13.5; 95% CI 1.3–147.6; *p =* 0.0348). Combined clinical decision-making utilizing factors such as GEP class, American Joint Committee on Cancer (AJCC) stage, and COMS size could have a significant clinical impact by improving risk stratification and adapting follow-up intervals in UM Class 1A patients.

## 1. Introduction

Uveal melanoma (UM) is the most common primary malignancy of the eye. The metastatic disease most commonly affects the liver and is fatal in approximately 50% of patients [1]. As the vast majority of patients present without evidence of metastatic disease at the time of diagnosis, stratification of metastatic risk can significantly influence the interval and length of surveillance. A prognostic molecular test based on gene expression profiling (GEP) has been proposed, validated prospectively, and is now in common use to assess metastatic risk [2,3,4,5]. The GEP assay uses a PCR-based 15-gene array to classify tumors as Class 1 (low metastatic risk) or Class 2 (high metastatic risk). While Class 1 tumors are typically associated with *EIF1AX* and *SF3B1* mutations, Class 2 tumors often present with *BAP1* mutations [6,7,8]. Within Class 1 tumors, expression profiles of *CDH1* and *RAB31* are used to further subdivide these tumors into Class 1A (low metastatic risk; 2% risk of metastasis at 3 and 5 years) and Class 1B (low–intermediate metastatic risk; 7% risk of metastasis at 3 years and 21% at 5 years). In contrast to Class 1A and 1B tumors, Class 2 tumors present with a significantly higher metastatic risk (50% risk of metastasis at 3 years and 72% at 5 years) [3]. Recently, expression of PRAME (Preferentially Expressed Antigen in Melanoma), a cancer-testis antigen, has been described as an independent biomarker for metastasis [9]. While PRAME-positive Class 1 UM patients might be at elevated risk for metastasis, PRAME positivity might correlate with a shorter time to metastasis in Class 2 UM patients [10,11].

We hypothesized that additional clinical and molecular features might help to identify UM patients with higher metastatic risk within the low-risk Class 1A group. To test this, we compared metastatic risk with several clinical features, tumor size and site, PRAME expression, *GNAQ* and *GNA11* mutation status, and lines of treatment in a cohort of 73 Class 1A UM patients.

## 2. Materials and Methods

### 2.1. Study Design and Objectives

This was a single-center IRB-approved retrospective case series. All UM patients with a Class 1A GEP signature from samples between February 2006 and March 2019 at MD Anderson Cancer Center, Houston, TX, were included. As part of routine clinical care, tumor samples were sent to Castle Biosciences (Friendswood, TX, USA) for analysis of GEP and, as of 2017, for PRAME status. Patient age at diagnosis, gender, American Joint Committee on Cancer (AJCC) 8th edition stage, primary tumor location, apical thickness, largest basal diameter (LBD), Collaborative Ocular Melanoma Study (COMS) size, mutation status (*GNAQ*, *GNA11*), primary therapy, time to distant metastasis (TTDM), time to distant or local recurrence (TTR), and overall survival (OS) were extracted and calculated from electronic medical records.

### 2.2. Statistics

Patient demographics, treatment, and clinical features were summarized using mean, median, SD, and minimum/maximum values for continuous variables and *n* (%) for categorical/ordinal variables. Patients were followed from the initial presentation date for primary UM until local recurrence for cumulative incidence (CI) of local recurrence, from initial presentation until the development of metastatic disease for CI of metastatic disease, and from presentation until death from any cause for OS. Patients not experiencing an outcome were censored at the date of last follow-up for local and distant recurrence and for OS.

OS and CI of distant metastasis were estimated using the Kaplan–Meier method, and differences between strata based on clinical characteristics were assessed using log-rank tests. Associations between clinical features and outcomes were evaluated using univariable and multivariable Cox regression analysis. Selection of variables for inclusion in the final multivariate models was based on having a likelihood ratio test *p*-value < 0.1. Time-dependent and integrated areas under the curve (AUC) were calculated to assess the predictive accuracy of each multivariate survival model.

## 3. Results

### 3.1. Patient Characteristics

A total of 73 patients were included in the analysis. Table 1 presents a summary of demographic and clinical characteristics overall and by local/distant recurrence status. The median age at initial presentation of primary UM in all patients was 60 years, and 57.5% (*n =* 42) of patients were female. Overall, 20.0% (*n =* 14) of patients were diagnosed with stage I disease, 65.7% (*n =* 46) were stage II, and the remaining 14.3% (*n =* 10) had stage III disease. Notably, patients who went on to develop distant metastases had a higher AJCC stage (IIIA, IIIB) compared to those who did not (*p =* 0.0116). The median apical thickness and LBD were 5.0 mm and 12.3 mm, respectively, in the total study population. A large majority of patients (*n =* 68; 93.2%) had a choroidal primary site and primary therapy distribution was 52.1% (*n =* 38) brachytherapy vs. 47.9% (*n =* 35) enucleation. Primary therapy was fairly evenly split between brachytherapy (*n =* 38; 52.1%) and enucleation (*n =* 35; 47.9%) in the overall population and differed significantly by recurrence status (*p =* 0.0497). An overwhelming majority (*n =* 6; 85.7%) of patients in the distant metastasis cohort underwent enucleation vs. only 45.3% (*n =* 29) of patients who did not experience any recurrence and no patients in the local recurrence cohort. Mutation status was missing for most patients as it does not play a role in primary UM management, and the tumor was unavailable for sequencing in the majority of cases. Of four patients with known mutations, 50% (*n =* 2) had the *GNAQ* mutation, and 50% (*n =* 2) had the *GNA11* mutation. PRAME status was available for 30.1% (*n =* 22), 31.8% (*n =* 7) of which were PRAME-positive.

### 3.2. Outcomes

Over a median follow-up of 37 months (range 1–119), seven (9.5%) patients developed distant metastasis, two (2.7%) patients developed primary recurrence, and eight died over the course of the study’s follow-up period. Of the eight patients who died, four died with metastatic disease while four died without. The median OS for the four patients who died without metastatic disease was 49 months (range 43–67 months), while the four who died with metastatic disease had a median survival of 36 months (range 13–50 months). The cause of death for three of the patients with metastatic disease was unknown, while the cause of death for one patient with metastatic disease was unrelated to UM.

Figure 1 illustrates the cumulative incidence (CI) of primary recurrence and distant metastasis separately as well as the combined CI of both outcomes. The overall 5-year CI was 21.1% (95% confidence interval (95% CI) 9.1–36.6%) for both outcomes. The 5-year CI of distant metastasis (17.0%; 95% CI 6.3–32.2%) was greater than the 5-year CI of local recurrence (4.2%; 95% CI 0.8–12.9%); however, this difference was not statistically significant (*p =* 0.0921). The overall study population’s 5-year OS was 79.2% (95% CI 60.2–89.8%). Of interest, patients who developed distant metastasis had a significantly shorter OS compared to patients who did not (*p =* 0.0002; Figure 2).

### 3.3. Predictors of Distant Metastasis Development and OS

Table 2 presents results of univariable Cox regression analysis of distant metastasis, local recurrence/distant metastasis, and overall survival. Factors associated with distant metastasis alone and combined local/distant recurrence included AJCC stage, apical thickness, LBD, and COMS size. As shown in Figure 3A, patients with stage III disease had a significantly higher CI of distant metastasis compared to stage I/II patients (*p* < 0.0001). Similarly, patients with a large COMS size had a significantly higher CI of distant metastasis than patients with a small or medium COMS size (*p* < 0.0001; Figure 3B). In 64 patients who did not develop local recurrence or distant metastases, LBD (mean *=* 12.2 mm, standard deviation (SD) 3.3 mm) and large COMS sizes (*n =* 12, 18.2%) were significantly lower than in seven patients who developed the distant metastatic disease (LBD: mean *=* 16.2 mm, SD 4.9 mm; large COMS size: *n =* 5, 71.4%; *p =* 0.0048, and *p =* 0.0018, respectively; Table 1).

In multivariable analyses of TTDM, TTR, and OS, we found that AJCC stage was significantly associated with TTDM, while primary therapy was significantly associated with OS (Figure 4). After controlling for other clinical factors, stage III disease was independently associated with a 20-fold increased risk of distant metastasis relative to stage I/II disease (HR 20.7; 95% CI 1.4–300.6; *p =* 0.0264). It should be noted that a higher stage was also associated with less favorable TTR and OS but with borderline statistical significance (multivariable *p =* 0.0659 for TTR and *p =* 0.0556 for OS). Enucleation was associated with a greater than 13.5-fold increased risk of mortality vs. brachytherapy (HR 13.5; 95% CI 1.3–147.6; *p =* 0.0335) after controlling for stage, apical thickness, LBD, and COMS size. Finally, the apical thickness was marginally, yet consistently, associated with TTDM (HR 1.6; *p =* 0.1008), TTR (HR 1.3; *p =* 0.1408), and OS (HR 1.6; *p =* 0.114) after controlling for other factors.

The integrated area under the curve (AUC) for the multivariate model for distant metastasis was 0.9618, with consistently high time-dependent AUCs at 12 months (AUC *=* 0.9466), 24 months (AUC *=* 0.9266), and 36 months (AUC *=* 0.9216) (Figure 5). The predictive accuracy of the multivariable model for OS was lower (integrated AUC *=* 0.8134) than that of the TTDM model.

## 4. Discussion

Multiple risk factors have been identified to be associated with recurrence and/or metastatic risk in UM. In line with recent study results, size parameters such as LBD and AJCC stage were significantly associated with GEP and with the development of metastasis [12,13,14]. The results of the present study suggest that Class 1A UM should be further stratified by size parameters such as AJCC stage, COMS size, or LBD to warrant a more accurate assessment of recurrence and/or metastatic risk. These additional stratifications are essentially overlapping criteria, all related to tumor size. Our findings are concordant with those of the Binkley et al. report, which identified the unique feature that Class 1A tumors that developed metastasis were also COMS large-sized tumors. Choice of primary therapy trended with the development of metastasis in our study, with patients who developed distant metastasis having a disproportionally larger amount of enucleations. The 1300-patient COMS medium-sized UM report noted equal rates of metastasis in patients who underwent brachytherapy compared to patients who underwent enucleation, ultimately disproving the Zimmerman hypothesis [15,16]. As such, not much should be made from our finding of the choice of primary therapy in Class 1A patients since our cohort contained a majority of COMS medium-sized tumors. Little is known about the impact of primary therapy on survival by GEP or by stage and whether adjuvant radiation may play a role in Class 1A UM patients with stage III or COMS large-sized tumors. 

The 5-year CI for development of distant metastasis in Class 1A tumors is 1–2% [3,4,5,13] but our study estimated a 5-year CI of distant metastasis of 17% for Class 1A patients, a finding that is aligned with the original GEP data that dichotomized patients into only two groups: Class 1 and Class 2. The subdividing of Class 1 tumors by the expression profiles of two of the 12 discriminating genes, *CDH1* and *RAB31*, is what further separates tumors into Class 1A and Class 1B. The Collaborative Ocular Oncology Group (COOG) Report Number 1 described GEP only in terms of Class 1 and Class 2 [3]. Our study highlights a single-institutional cohort of Class 1A UM patients who are estimated to develop metastasis as a rate of 17% at 5 years with specific risk factors (AJCC stage III disease) compared to Class 1A UM patients who do not go on to develop metastatic disease at 5 years (AJCC stage I/II disease). It was this clinical finding of a higher rate of metastasis in our Class 1A patients that caught our attention and prompted us to validate this observation. 

The median OS of patients in our cohort with Class 1A metastatic disease was 36 months (13–50 months) and the 5-year OS of the overall Class 1A study population was 79.2%. A COMS report of 5-year estimates of survival was 84% for small tumors, 68% for medium tumors, and 47% for large tumors [17]. From the Helsinki University Central Hospital, 62% of patients who died of metastatic UM did so within 5 years [18], and a Surveillance, Epidemiology, and End Results (SEER) database report noted a mean 5-year UM-specific survival rate of 76% [19]. COOG and other reports of GEP describe metastasis-free survival but do not describe OS. 

In previous publications, PRAME expression in UM was found to be associated with increasing LBD and higher GEP class [10,11]. Curiously, nearly a third of Class 1A UM in one report were found to express PRAME, yet Class 1A tumors are reported to carry a low metastatic risk of 2% at 5 years. Therefore, the role of PRAME expression is not yet solidified, particularly in low GEP tumors. Hence, survival follow-up should be extended well past 5 years to determine when PRAME-positive Class 1 tumors go on to develop metastasis. PRAME status was captured in merely 30% of patients in our cohort. Because PRAME testing has only been available for the past few years, the majority of our patient cohort were analyzed in the era before the routine use of PRAME expression. Thus, the significance of PRAME expression should be evaluated in a larger cohort of Class 1A patients using multivariate models of risk factors. 

The presence of driver mutations such as *BAP1* and *SF3B1* has been shown to be strongly associated with metastasis and melanoma-specific mortality independent of GEP class [20,21]. The mutually exclusive, highly conserved mutations in *GNAQ* or *GNA11* were not found to be prognostically significant in a previous study [20]. It should be noted that the impact of *GNAQ* or *GNA11* mutations on recurrence risk was limited in our study due to the low number of patients evaluated for these mutations. Mutation status was missing for most patients in our study, as it does not play a role in primary tumor management, and the tumor was unavailable for DNA sequencing due to the choice of primary therapy in the majority of cases. A future or larger cohort including sufficient PRAME and *GNAQ/GNA11* testing could be interrogated in a similar fashion using univariate and multivariate analysis of risk factors.

The National Comprehensive Cancer Network (NCCN) has incorporated additional risk factors other than GEP into their UM consensus guidelines [22,23]. Surveillance is stratified by risk assessment for distant metastasis and includes consideration of: GEP, chromosome 3, 6, and 8 status, mutation status, PRAME expression, and AJCC T-category. Imaging is recommended annually for low-risk patients, every 6–12 months for 10 years for medium-risk patients, and every 3–6 months for 5 years then every 6–12 months for years 6–10 for high-risk patients. The findings from our cohort study would bump surveillance of certain Class 1A patients from once a year if solely considering GEP to every 3–6 months if considering GEP and AJCC T-category.

Our study had several limitations. The study was limited by its retrospective, single-center design. Moreover, to smooth our analysis of anatomic location, it was necessary to collapse tumors with overlapping anatomic sites as follows: ciliochoroidal tumors were classified as choroidal, and iridociliochoroidal tumors were classified as the ciliary body. Despite these limitations, our study demonstrated the impact of size parameters such as AJCC stage and tumor size for improving metastatic risk assessment of Class 1A tumors in UM, which is line with previous findings that tumor size plays a role in the prognostication of low-risk UM [12,24].

## 5. Conclusions

Class 1A UM patients with stage III disease are at elevated risk for metastasis. Combined clinical decision-making utilizing AJCC stage could have a significant clinical impact by improving risk stratification and adapting follow-up intervals in Class 1A UM patients.

## Figures and Tables

**Figure 1 cancers-13-03292-f001:**
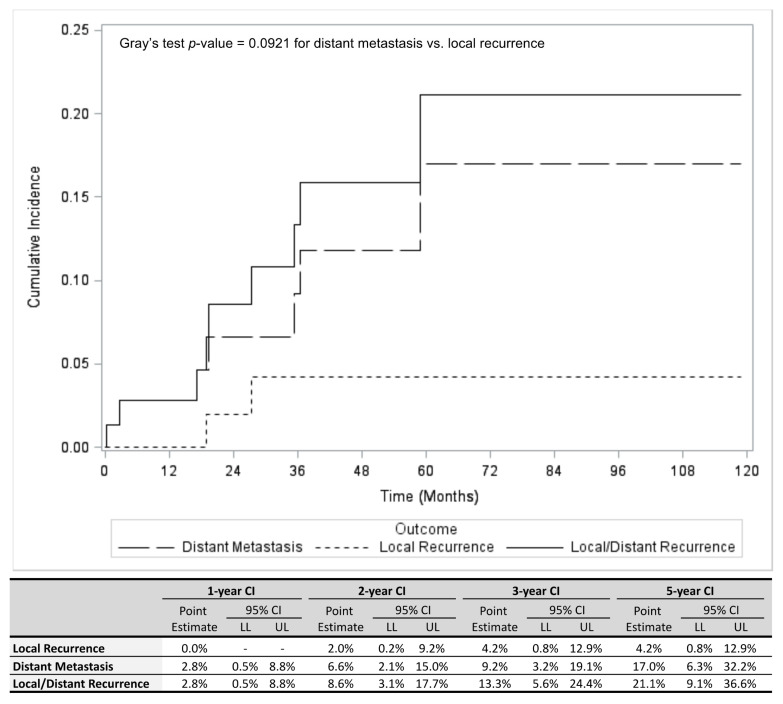
Cumulative incidence (CI) of distant metastasis, local recurrence, and combined distant metastasis and local recurrence is shown (top panel), including estimates for 1, 2, 3, and 5 years (bottom panel) in patients with Class 1A UM. Abbreviations: HR, Hazard ratio; LL, Lower limit; UL, Upper Limit; CI, Confidence interval; Cumulative incidence.

**Figure 2 cancers-13-03292-f002:**
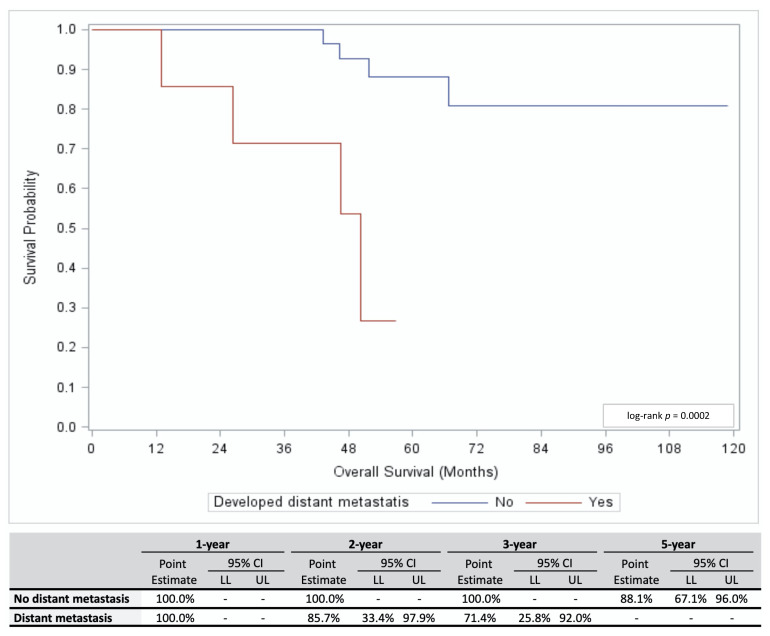
Overall survival for Class 1A UM patients who did (red line) or did not (blue line) develop distant metastasis is shown (top panel), including estimates for 1, 2, 3, and 5 years (bottom panel). Abbreviations: HR, Hazard ratio; LL, Lower limit; UL, Upper Limit; CI, Confidence interval.

**Figure 3 cancers-13-03292-f003:**
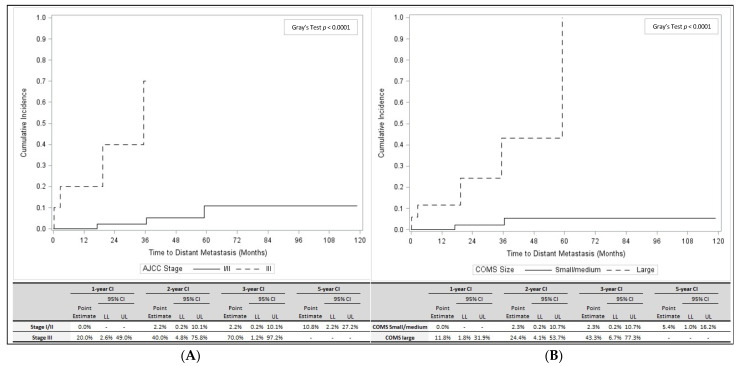
(**A**)**.** Cumulative incidence of distant metastasis for patients with AJCC stage I/II disease (solid line) and AJCC stage III disease (dashed line) is shown (top panel), including estimates for 1, 2, 3, and 5 years (bottom panel). (**B**)**.** Cumulative incidence of distant metastasis for patients with small/medium COMS size (solid line) and large COMS size (dashed line) is shown (top panel), including estimates for 1, 2, 3, and 5 years (bottom panel). Abbreviations: COMS, Collaborative Ocular Melanoma Study; HR, Hazard ratio; LL, Lower limit; UL, Upper Limit; CI, Confidence interval.

**Figure 4 cancers-13-03292-f004:**
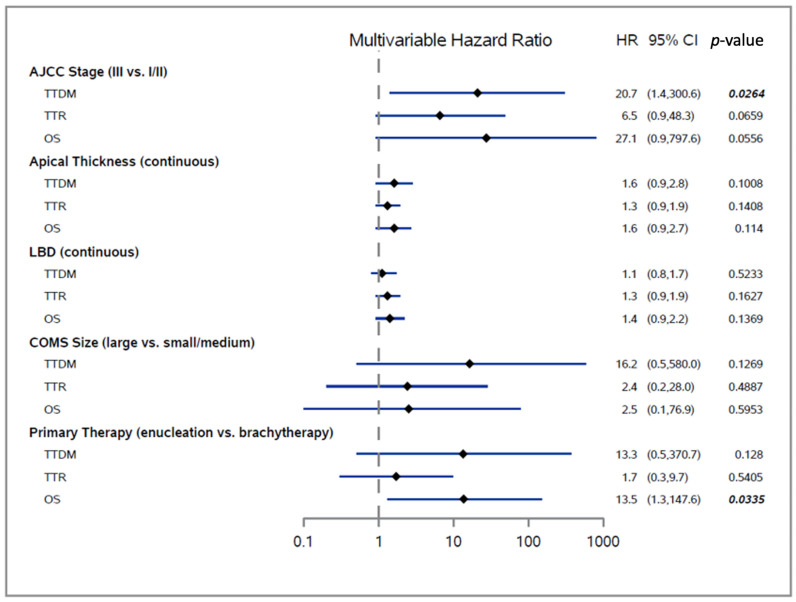
Forest plot of TTDM, TTR, and OS. Significant *p*-values are shown in bold and italic. Abbreviations: TTDM, Time to distant metastasis; TTR, Time to local or distant recurrence; OS, Overall survival; HR, Hazard ratio; CI, Confidence interval.

**Figure 5 cancers-13-03292-f005:**
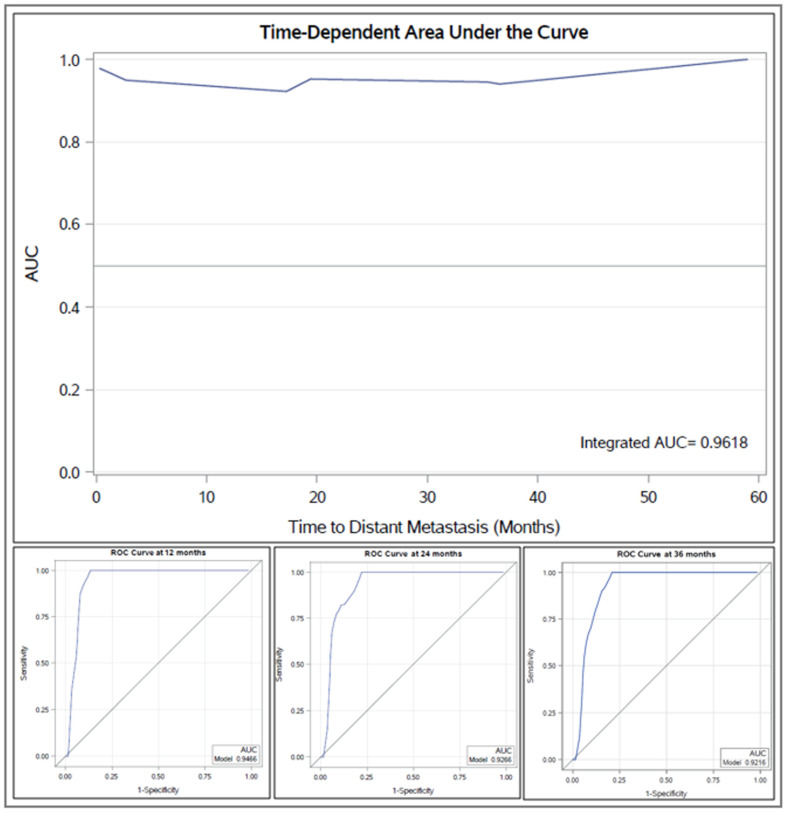
Time-dependent AUC for receiver operating characteristics (ROC) of multivariate model for distant metastasis is shown (top panel), including ROC curves at 1, 2, and 3 years (bottom panel). Abbreviations: AUC, Area under the curve; ROC, Receiver operating characteristics.

**Table 1 cancers-13-03292-t001:** Patient characteristics.

Variable	All Patients (*n =* 73)		No Local Recurrence or Distant Metastases (*n =* 64)		Local Recurrence (*n =* 2)		Distant Metastasis (*n =* 7)		*p*-Value * (Distant Metastasis Yes vs. No)
	*n*	%	*n*	%	*n*	%	*n*	%	
Age at Initial Diagnosis									0.0622
Mean (Standard Deviation)	59.6 (11.3)		58.9 (11.6)		70.9 (18.2)		62.5 (2.3)		
Median (Minimum, Maximum)	61 (32, 86)		59 (32, 86)		71 (58, 84)		62 (59, 65)		
Gender									0.4484
Female	42	57.5%	38	59.4%	1	50.0%	3	42.9%	
Male	31	42.5%	26	40.6%	1	50.0%	4	57.1%	
AJCC Stage									0.0116
I	14	20.0%	13	21.3%	0	0.0%	1	14.3%	
IIA	27	38.6%	24	39.3%	2	100.0%	1	14.3%	
IIB	19	27.1%	18	29.5%	0	0.0%	1	14.3%	
IIIA	7	10.0%	5	8.2%	0	0.0%	2	28.6%	
IIIB	3	4.3%	1	1.6%	0	0.0%	2	28.6%	
Apical Thickness (mm)									0.2446
Mean (Standard Deviation)	5.9 (3.2)		5.7 (2.9)		4.3 (1.3)		8.2 (5.2)		
Median (Minimum, Maximum)	5 (2.0, 15.7)		4.6 (2.3, 14.6)		4.3 (3.3, 5.2)		8.5 (2.0, 15.7)		
LBD (mm)									0.0048
Mean (Standard Deviation)	12.6 (3.6)		12.2 (3.3)		13.3 (0.9)		16.2 (4.9)		
Median (Minimum, Maximum)	12.3 (4.5, 21.0)		11.9 (4.5, 21.0)		13.3 (12.6, 13.9)		17.6 (7.0, 21.0)		
COMS Size									0.0018
Small	3	4.1%	2	3.0%	0	0.0%	1	14.3%	
Medium	53	72.6%	52	78.8%	2	100.0%	1	14.3%	
Large	17	23.3%	12	18.2%	0	0.0%	5	71.4%	
PRAME									0.3182
Positive	7	9.6%	6	9.4%	0	0.0%	1	14.3%	
Negative	15	20.5%	15	23.4%	0	0.0%	0	0.0%	
Not tested	51	69.9%	43	67.2%	2	100.0%	6	85.7%	
Mutation									1.0000
*GNAQ*	2	2.7%	1	1.6%	0	0.0%	1	14.3%	
*GNA11*	2	2.7%	0	0.0%	0	0.0%	2	28.6%	
Not Tested	69	94.5%	63	98.4%	2	100.%	4	57.1%	
Primary Site									0.405
Choroidal	68	93.2%	60	93.8%	2	100.0%	6	85.7%	
Ciliary body	5	6.8%	4	6.3%	0	0.0%	1	14.3%	
Primary Therapy									0.0497
Brachytherapy	38	52.1%	35	54.7%	2	100.0%	1	14.3%	
Enucleation	35	47.9%	29	45.3%	0	0.0%	6	85.7%	

* Based on *t*-tests/Wilcoxon rank sum tests for continuous variables and chi-square/Fisher’s exact tests for categorical variables. Abbreviations: AJCC, American Joint Committee on Cancer; LBD, Largest basal tumor diameter, COMS, Collaborative Ocular Melanoma Study.

**Table 2 cancers-13-03292-t002:** Univariable analysis of outcomes.

Variable	Distant Metastasis				Local/Distant Recurrence				Overall Survival			
	95% CI				95% CI				95% CI			
	HR	LL	UL	*p*-Value	HR	LL	UL	*p*-Value	HR	LL	UL	*p*-Value
Diagnosis Age (continuous)	1.02	0.96	1.09	0.5101	1.04	0.98	1.11	0.1952	1.04	0.98	1.11	0.2151
Gender (male vs. female)	2.07	0.46	9.27	0.3419	1.92	0.52	7.16	0.3315	3.20	0.76	13.52	0.1129
AJCC Stage (III vs. I/II)	24.26	4.17	141.15	0.0004	11.99	2.87	50.04	0.0007	7.64	1.25	46.83	0.0280
Apical Thickness (continuous)	1.30	1.08	1.57	0.0057	1.22	1.03	1.45	0.0203	1.15	0.95	1.39	0.1506
LBD (continuous)	1.49	1.15	1.94	0.0026	1.39	1.12	1.72	0.0026	1.21	0.97	1.50	0.0889
COMS Size (large vs. small/medium)	16.95	3.12	92.02	0.0010	7.97	2.07	30.71	0.0026	2.76	0.52	14.67	0.2329
Primary Site (ciliary body vs. choroidal)	2.54	0.31	21.13	0.3885	1.92	0.24	15.40	0.5377	0.00	0.00	-	0.9949
Primary Therapy (enucleation vs. brachytherapy)	7.59	0.91	63.34	0.0610	2.44	0.61	9.79	0.2076	9.16	1.12	74.58	0.0385

## Data Availability

The data that support the findings of this study are available from the corresponding author upon reasonable request.

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
