# Peer review of "Metastatic Risk Factors Associated with Class 1A Uveal Melanoma Patients"

_cancers, 2021, doi:10.3390/cancers13133292_

Round 1

Reviewer 1 Report

Excellent manuscript. Few additions/comments.

  1. The paragraph "This brings back memories of the Zimmerman 195 hypothesis, which suggested an iatrogenic cause of distant metastasis due to tumor seeding from an enucleation procedure" can mislead the readers, even though the following sentence disregards it. Please remove it from the manuscript
  2. Add the reference and a comment on:     Do Largest Basal Tumor Diameter and the American Joint Committee on Cancer's Cancer Staging Influence Prognostication by Gene Expression Profiling in Choroidal Melanoma. Demirci H, Niziol LM, Ozkurt Z, Slimani N, Ozgonul C, Liu T, Musch DC, Materin M.Am J Ophthalmol. 2018 Nov;195:83-92. doi: 10.1016/j.ajo.2018.07.033. Epub 2018 Aug 4.PMID: 30081017  

Author Response

Reviewer #1:

Reviewer comment #1: The paragraph "This brings back memories of the Zimmerman 195 hypothesis, which suggested an iatrogenic cause of distant metastasis due to tumor seeding from an enucleation procedure" can mislead the readers, even though the following sentence disregards it. Please remove it from the manuscript

AU REPLY: We agree that the sentence could be misleading and have removed it from the manuscript.

Reviewer comment #2: Add the reference and a comment on: Do Largest Basal Tumor Diameter and the American Joint Committee on Cancer's Cancer Staging Influence Prognostication by Gene Expression Profiling in Choroidal Melanoma. Demirci H, Niziol LM, Ozkurt Z, Slimani N, Ozgonul C, Liu T, Musch DC, Materin M.Am J Ophthalmol. 2018 Nov;195:83-92. doi: 10.1016/j.ajo.2018.07.033. Epub 2018 Aug 4.PMID: 30081017  

AU REPLY: This reference has been added to the Discussion, lines 190-191.

Reviewer 2 Report

Ballhausen et al present a single institution retrospective study of a relatively large cohort of class 1A uveal melanoma patients to determine additional risk factors that may contribute to distant metastatic recurrence. The major finding is that tumor size/depth (LBD, AJCC, COMS) are prognostic predictors of distant metastasis. While this is not a novel finding, as it has been reported previously that AJCC/LBD/COMS are independent predictors than GEP, this report offers additional value in that surveillance recommendations of uveal melanoma is based on GEP alone. While the distant metastasis risk of class 1A patients is low, it is not 0%. Thus, identifying additional factors that could predict patients with higher recurrence risk, can fine tune recommendations of surveillance imaging - increasing the frequency and chance of early detection in high risk while minimizing unnecessary scans in low risk. Furthermore, the authors identify a very intriguing hypothesis-generating observation of the impact of primary treatment and stage on recurrence risk.

Recommendations include:

  1. Table 1
    1. Local recurrence and Distant metastasis are duplicated in COMS size. Please revise.
    2. The reporting of GNAQ GNA11 mutation frequency is misleading. The frequency is not 1-15%. It is that there is a lot of missing information. I would remove this from the table given the majority of data is missing. The description in the text is sufficient.
    3. PRAME status should be added to the table (positive, negative, missing).
    4. In results 3.1, line 105 and line 111 both discuss primary treatment but are split by discussion on GNA mutation status. Please reorganize to improve flow.
  2. Figure 1
    1. The CI of distant metastasis for class 1A is expected to be 2% by 5 years. In this cohort, it is almost 20%. This should be addressed somewhere in the manuscript.
    2. Did the 4 patients who died without distant disease die of disease-related causes?
    3. How does OS of this cohort compare to historical. Similar to above, it is important to put the patient population of this cohort into context with other published cohorts - are they similar? are the different? what are key differences?
  3. Figure 2. The legend should be DM status not TTDM status.
  4. Discussion.
    1. The discussion would be greatly strengthened by expanding on the potential hypothesis that enucleation may be less effective than brachytherapy for higher stage class 1A UM. The Zimmerman hypothesis is interesting but maybe it is not a seeding problem versus a local control problem in general. Regardless, this observation raises interesting questions regarding which therapy is more effective - what is known in the literature comparing these two modalities to each other based on GEP and stage? Should we be recommending adjuvant RT to all patients undergoing enucleation and found to have high risk features? 
    2. Similarly, a discussion on the implications of additional risk factors other than GEP on surveillance intervals should be mentioned as this has important clinical impact.
  5. References. Please carefully review that all references in the text are appropriately aligned with the reference list. For example, in the discussion, the second paragraph references #5 for PRAME but should be #6. Likewise, the third paragraph references #6 for GNA but is inaccurate. The whole manuscript should be reviewed to ensure accuracy.

Minor grammatical points:

  1. Line 85. Remove "or death" as death is an outcome so would not apply to the referenced "Patients not experiencing an outcome..."
  2. Line 90. Punctuation is missing between "analysis" and "Selection".
  3. Line 120. Remove "the" before "metastatic disease".
  4. Line 212. "and" should not be italicized.

Author Response

Reviewer #2

Reviewer comment #1.1 (Table 1): Local recurrence and Distant metastasis are duplicated in COMS size. Please revise.

AU REPLY: Values for COMS size of Local recurrence and Distant metastasis cohort have been corrected in Table 1.

Reviewer comment #1.2 (Table 1): The reporting of GNAQ GNA11 mutation frequency is misleading. The frequency is not 1-15%. It is that there is a lot of missing information. I would remove this from the table given the majority of data is missing. The description in the text is sufficient.

AU REPLY: Most cases of our cohort did not have GNAQ and GNA11 mutation status or PRAME status as the patients were treated before these parameters came into routine clinical testing. We have added PRAME status to Table 1 as suggested in comment 1.3. Due to the amount of missing data for both mutation status and PRAME status we would suggest to either keep or alternatively remove both variables in Table 1 at the journal’s discretion. As a reference piece, it would make sense to keep both items but note the significant amount of missing data due to the era in which these cases were diagnosed and treated (addressed in the text).

Reviewer comment #1.3 (Table 1): PRAME status should be added to the table (positive, negative, missing).

AU REPLY: PRAME status has been added to Table 1 as suggested.

Reviewer comment #1.4: In results 3.1, line 105 and line 111 both discuss primary treatment but are split by discussion on GNA mutation status. Please reorganize to improve flow.

AU REPLY: Thank you for this helpful comment. The section on mutation and PRAME status has been moved to the end of Results 3.1 (lines 111-115).

Reviewer comment #2.1 (Figure 1): The CI of distant metastasis for class 1A is expected to be 2% by 5 years. In this cohort, it is almost 20%. This should be addressed somewhere in the manuscript.

AU REPLY: Thank you for this request. This has been addressed in the Discussion (lines 207-208).

Reviewer comment #2.2: Did the 4 patients who died without distant disease die of disease-related causes?

AU REPLY: Of 4 patients who died without distant disease, 1 patient died of disease-unrelated causes. For remaining 3 patients, cause of death was not available. This information has been added to Results, Section 3.2 (lines 125-12).

Reviewer comment #2.3: How does OS of this cohort compare to historical. Similar to above, it is important to put the patient population of this cohort into context with other published cohorts - are they similar? are the different? what are key differences?

AU REPLY: A description of OS from other uveal melanoma reports has been added to the Discussion (lines 220-227)

Reviewer comment #3.1 (Figure 2): The legend should be DM status not TTDM status.

AU REPLY: Thank you for this typo. The legend of Figure 2 has been changed to DM status instead of TTDM status.

Reviewer comment #4.1: The discussion would be greatly strengthened by expanding on the potential hypothesis that enucleation may be less effective than brachytherapy for higher stage class 1A UM. The Zimmerman hypothesis is interesting but maybe it is not a seeding problem versus a local control problem in general. Regardless, this observation raises interesting questions regarding which therapy is more effective - what is known in the literature comparing these two modalities to each other based on GEP and stage? Should we be recommending adjuvant RT to all patients undergoing enucleation and found to have high risk features? 

AU REPLY: This is a very interesting point where little is known about choice of primary therapy as it related to outcomes by GEP and Stage and the utility of adjuvant RT. We have added commentary on this point (lines 202-206).

Reviewer comment #4.2: Similarly, a discussion on the implications of additional risk factors other than GEP on surveillance intervals should be mentioned as this has important clinical impact.

AU REPLY: Implications of metastatic risk factors on surveillance intervals of uveal melanoma patients are now discussed in an additional paragraph (lines 251-258).

Reviewer comment #5 (References): Please carefully review that all references in the text are appropriately aligned with the reference list. For example, in the discussion, the second paragraph references #5 for PRAME but should be #6. Likewise, the third paragraph references #6 for GNA but is inaccurate. The whole manuscript should be reviewed to ensure accuracy.

AU REPLY: References have been carefully reviewed and corrected to ensure accuracy.

Reviewer comment #6.1-#6.4 (Minor grammatical points):

  1. Line 85. Remove "or death" as death is an outcome so would not apply to the referenced "Patients not experiencing an outcome..."
  2. Line 90. Punctuation is missing between "analysis" and "Selection".
  3. Line 120. Remove "the" before "metastatic disease".
  4. Line 212. "and" should not be italicized.

AU REPLY: Many thanks for pointing this out. The grammatical errors have been corrected in the revised version along with some additional other minor typographic edits.